# PAPPI: Personalized analysis of plantar pressure images using statistical modelling and parametric mapping

Brian G. Booth[1]*, Eva Hoefnagels[2], Toon Huysmans[1,3], Jan Sijbers[1], Noël L. W. Keijsers[4]

**1** imec-Vision Lab, Department of Physics, University of Antwerp, Antwerp, Belgium, **2** Department of Orthopaedics, Sint Maartenskliniek, Woerden, The Netherlands, **3** Section on Applied Ergonomics & Design, Faculty of Industrial Design Engineering, Delft University of Technology, Delft, The Netherlands, **4** Department of Research, Sint Maartenskliniek, Nijmegen, The Netherlands

* brian.booth@uantwerpen.be

**Data Availability Statement:** The datasets used in this study are available online at Zenodo. The healthy control dataset is available at https://doi.org/10.5281/zenodo.1265419, while the Hallux

## Abstract

Quantitative analyses of plantar pressure images typically occur at the group level and under the assumption that individuals within each group display homogeneous pressure patterns. When this assumption does not hold, a personalized analysis technique is required. Yet, existing personalized plantar pressure analysis techniques work at the image level, leading to results that can be unintuitive and difficult to interpret. To address these limitations, we introduce PAPPI: the Personalized Analysis of Plantar Pressure Images. PAPPI is built around the statistical modelling of the relationship between plantar pressures in healthy controls and their demographic characteristics. This statistical model then serves as the healthy baseline to which an individual's real plantar pressures are compared using statistical parametric mapping. As a proof-of-concept, we evaluated PAPPI on a cohort of 50 hallux valgus patients. PAPPI showed that plantar pressures from hallux valgus patients did not have a single, homogeneous pattern, but instead, 5 abnormal pressure patterns were observed in sections of this population. When comparing these patterns to foot pain scores (i.e. Foot Function Index, Manchester-Oxford Foot Questionnaire) and radiographic hallux angle measurements, we observed that patients with increased pressure under metatarsal 1 reported less foot pain than other patients in the cohort, while patients with abnormal pressures in the heel showed more severe hallux valgus angles and more foot pain. Also, incidences of pes planus were higher in our hallux valgus cohort compared to the modelled healthy controls. PAPPI helped to clarify recent discrepancies in group-level plantar pressure studies and showed its unique ability to produce quantitative, interpretable, and personalized analyses for plantar pressure images.

## Introduction

For gait-related complaints, plantar pressures—pressures between one's foot and the ground—have shown to be useful quantities to measure and analyze for diagnostic purposes [1]. In

Valgus dataset is available at https://doi.org/10.5281/zenodo.1441308.

**Funding:** B.G.B. received funding from the European Union's Horizon 2020 research and innovation programme under the Marie Sklodowska-Curie grant agreement no. 746614 (https://cordis.europa.eu/project/rcn/209771/factsheet/en). B.G.B., T.H., and J.S. received funding from imec Belgium through ICON grant no. 150218. (https://www.imec-int.com/en/what-we-offer/research-portfolio/footwork). The funders had no role in study design, data collection and analysis, decision to publish, or preparation of the manuscript.

**Competing interests:** The authors have declared that no competing interests exist.

recent years, the standardization of plantar pressure measurement devices [2] and the introduction of a variety of analysis techniques [3] have shown that there is a desire to capitalize on the information that these measurements provide. Despite these advancements, recent studies have shown that existing analysis techniques have limitations on what information can be extracted from plantar pressures [4] as well as the repeatability of studies involving that information [5].

From a quantitative and statistical perspective, the majority of plantar pressure analysis techniques operate at the group level [3, 6, 7]. In particular, statistical parametric mapping (SPM) techniques have recently gained popularity in the performance of these group studies, whether they be for region-of-interest studies [8], centre of pressure trajectories [9], pressure pattern images [10], or plantar pressure videos [11]. SPM works by bringing all plantar pressure measurements into anatomical (and possibly also temporal) alignment, then performing statistical tests at each sampled point (e.g. each pixel in an image or each time point in a sequence). By performing statistics in this manner, SPM localizes and highlights regions of plantar pressures that show statistically significant group differences, thereby simplifying the interpretation of a study's results.

The motivation behind SPM group studies is to highlight abnormal pressure patterns that can be used as a biomarker for a particular foot complaint. However, the pressure patterns highlighted by these group-level statistical tests are only the ones that consistently differ between the groups, where consistency is defined by the test's significance level (e.g. $\alpha = 0.05$ implying 95% of the time). As a result, the way groups are defined becomes a key and challenging parameter to set. Liberal inclusion and exclusion criteria can lead to large within-group variances, making it hard to observe group differences. Conversely, strict criteria can result in low group sizes, when in turn reduces the statistical power of the analysis.

One example of the sensitivity to group definitions in plantar pressure studies is the discrepancy in results reported on hallux valgus patients [5]. Table 1 summarizes the plantar pressure studies involving hallux valgus patients based on where group-level statistical tests showed significant differences. Note that none of the studies match exactly and, in some cases, they contradict each other. In particular, the works of Booth et al. and Galica et al. disagree about pressures under the lesser toes [5, 12]. Those two works also disagree with the works of Bryant et al., Hida et al., and Koller et al. regarding pressures under lateral forefoot [13–15].

The inconclusive results in these studies are not surprising. It is known that even healthy individuals show significant differences in plantar pressures [17]. There are also a variety of demographic factors that can impact plantar pressure measurements [18, 19]. While some of these factors can be statistically modelled as covariates, others might be unknown or not easy

**Table 1. Summary of the results of group-level plantar pressure analyses of hallux valgus patients.**

| Study | Location of Pressure Differences | | | | | |
|---|---|---|---|---|---|---|
| | **Heel** | **Midfoot** | **MT 1-2** | **MT 3-5** | **Hallux** | **Toes 2-5** |
| Booth et al. [5] | ↓ | | | ↓ | | ↓ |
| Bryant et al. [13] | | | ↑ | ↑ | | |
| Galica et al. [12] | ↓ | | | ↓ | ↓ | ↑ |
| Hida et al. [14] | | | | ↑ | | |
| Koller et al. [15] | | ↑ | | ↑ | ↓ | |
| Wen et al. [16] | | | ↑ | | ↓ | |

Up arrows indicate that the hallux valgus group had higher plantar pressures than healthy controls, while down arrows indicate the opposite. MT1 to MT5 refer to metatarsals 1-5. Note the lack of agreement between studies. See text for further details.

to model. These latter factors introduce within-group variance, making it not only harder to identify group-level statistical differences, but also making the group-level analysis sensitive to the specific individuals included in the study.

As a result of these limitations in group-level studies, cluster-based analysis techniques have begun to appear in plantar pressure studies [20–24]. These cluster analyses allow us to group plantar pressure measurements into distinctive clusters such that all individuals in a cluster show similar plantar pressures. While this analysis technique introduces the challenge of accurately choosing a priori the expected number of clusters, all these studies have been able to highlight that both patient and healthy control groups comprise of multiple clusters. These analyses also show that an individual's plantar pressures can differ as much between the clusters as they do between the patient groups. Overall, these results highlight the challenge in defining biomarkers of foot complaints based on plantar pressure measurements.

More recently, machine learning algorithms have been employed in order to classify an individual's plantar pressure measurement into patient or healthy control groups [25–29]. In these studies, a database of plantar pressure data is combined with the corresponding group memberships in order to define a non-linear regression function between the two quantities. A variety of machine learning algorithms have been used to perform this regression, from artificial neural networks [25, 27], to logistic regression [29], to nearest neighbour classification [28], to support vector machines [26]. Regardless of the algorithm used, the resulting classifier produces a personalized result: an individual's plantar pressure measurement, as a whole, gets labelled as either healthy or unhealthy. However, these machine learning techniques often appear to users as a black box: it is unclear how the algorithm is making its choice [30]. Specifically, classifiers have traditionally labelled the whole plantar pressure measurement, making it a challenge to localize the aspect of a person's gait underlying the classification result.

Despite all the advances in the analysis of plantar pressure measurements, there remains a need for an analysis technique that provides both a personalized analysis of a person's plantar pressures while also localizing abnormal pressure measurements to precise locations on the foot. The objective of this paper is to fill that gap with PAPPI: the Personalized Analysis of Plantar Pressure Images. Fig 1 shows how PAPPI relates to other plantar pressure analysis techniques. At a high level, PAPPI combines the localization benefits of SPM with the personalization benefits of classification algorithms. Like SPM, PAPPI is based on the idea of bringing plantar pressure measurements into anatomical alignment, then performing statistics at each pixel. Unlike SPM, PAPPI employs a statistical outlier detection algorithm to classify plantar pressure abnormalities pixel-by-pixel [31, 32]. This outlier detection involves the pixel-by-pixel modelling of plantar pressures from a healthy population as well as the relationship between those pressures and demographic factors such as age, weight, and gender. This model then serves as a healthy baseline to which an individual's plantar pressures are compared, pixel-by-pixel, using single-sample t-tests. Plantar pressures that do not agree with the model are then classified as abnormalities and are highlighted for display.

With PAPPI, we introduce two methodological contributions. First, we introduce pixel-by-pixel outlier detection to the analysis of plantar pressure measurements. Second, we incorporate the impact of multiple demographic factors into the statistical outlier detection, thereby allowing PAPPI to fine tune its outlier detection to specific individuals. With these contributions, we aim to provide the intuitive summaries of abnormal plantar pressures that SPM is known for, while also personalizing the SPM procedure in a way that accounts for the natural variability in plantar pressure measurements. As a proof-of-concept, we apply PAPPI to a cohort of hallux valgus patients and aim to show that a personalized analysis technique like PAPPI can clarify results from earlier hallux valgus group studies.

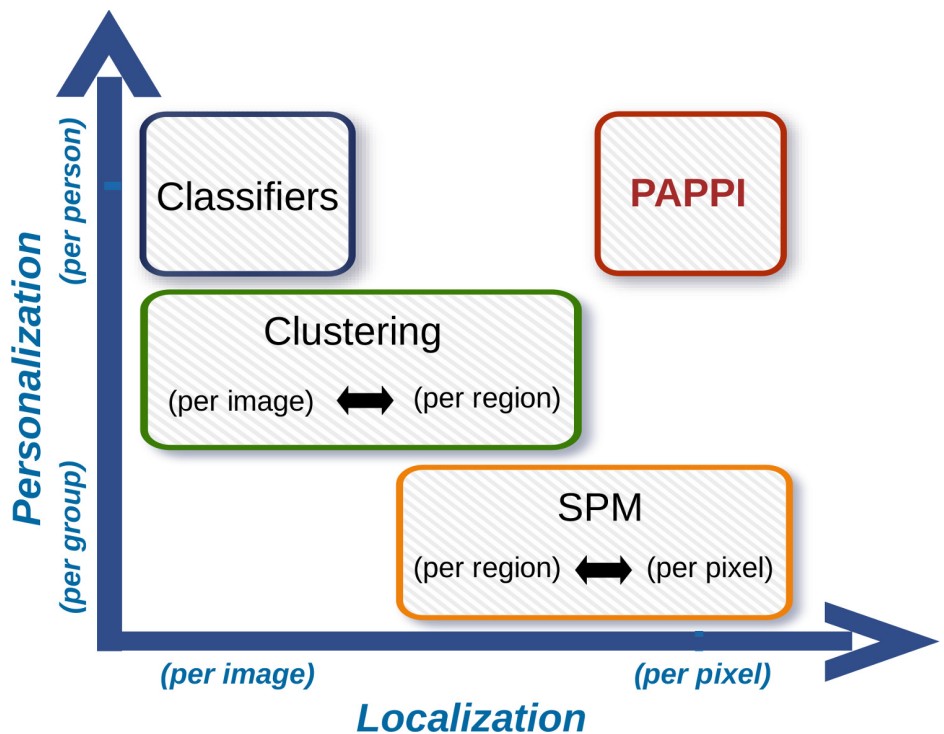

**Fig 1. The positioning of PAPPI with respect to competing plantar pressure analysis techniques.** The proposed PAPPI technique is unique in providing a highly personalized result (i.e. one result per foot) while also localizing plantar pressure abnormalities to a high degree (i.e. pixel-by-pixel abnormality identification).

## Materials and methods

### Data collection

To perform a proof-of-concept of PAPPI, we make use of dynamic plantar pressure measurements from the CAD WALK open access database [33]. Two data sets are employed here. First, plantar pressure measurements from 55 healthy controls are used to build the statistical model [34]. Second, plantar pressure measurements from 50 hallux valgus patients are used to evaluate the proposed PAPPI framework [35]. To our knowledge, no previous results have been reported on these datasets, so no multiple comparison issues exist between our study and previous works. In our descriptions, we will refer to a dynamic plantar pressure measurement as being a video $V$ containing plantar pressure samples indexed by a pixel location $\mathbf{x} = [x, y]$ and a time frame $t$.

The plantar pressure measurements were collected in two ways: (a) using an internally-calibrated 1.5 m footscan$^{\circledR}$ plate (rs scan, Paal, Belgium; dimensions: 160.5 × 46.9 cm, sensor dimensions: 0.762 × 0.508 cm), and (b) using a 0.5 m footscan$^{\circledR}$ plate (rs scan, Paal, Belgium; dimensions: 48.8 × 32.5 cm, sensor dimensions: 0.762 × 0.508 cm) on top of a Kistler force plate (9286AA, Kistler, Wintherthur, Switzerland), with both synchronized to each other using a rs scan footscan$^{\circledR}$ 3D interface box. The pressure data was gathered in rs scan's footscan$^{\circledR}$ software 7 gait 2$^{nd}$ generation, from which it was exported and then converted to NIfTI format using MATLAB version 2016b (The MathWorks, Natuck, USA). The plantar pressures of the healthy controls were measured 24 times per foot at a frequency of 500 Hz, while the hallux valgus patients were measured a minimum of 8 times per foot at a frequency of 200 Hz. Given the international standards for pressure-sensing plates [2], the difference in sampling

frequency is not expected to influence the accuracy of the measurements. All participants were measured using the 3-step protocol [36] while walking barefoot at their preferred walking speed.

In addition to the plantar pressure measurements, each participant's age, sex, shoe size, weight, and height were recorded. The participant's weight was measured using a traditional scale while the other measures are self-reported. These five demographic factors are collected in order to account for their effects on the plantar pressure measurements in the statistical modelling [18, 19].

For the hallux valgus group, additional clinical information was collected. First, the hallux valgus patients filled out two foot function self-assessment questionnaires: the 5 pt. Foot Function Index (FFI-5pt) [37] and the Manchester-Oxford Foot Questionnaire (MOXFQ) [38]. Additionally, the hallux valgus angle (HVA) and the intermetatarsal angle (IMA) of each patient were also recorded based on measurements from the patient's dorsoplantar weight-bearing radiograph. A patient's foot was deemed to have a hallux valgus if its IMA met or exceeded 9 degrees, or if its HVA met or exceeded 15 degrees [39]. From the 50 patients in our cohort, 69 hallux valgus cases met these inclusion criteria.

## Data preprocessing

The rs scan footscan® pressure plate used in the data collection has non-square sensor dimensions, resulting in the plantar pressure measurements being compressed in the anterior-posterior direction. In order to recover the original foot geometry, each plantar pressure measurement was upsampled to a 3 mm × 3 mm grid using cubic interpolation [11]. Each plantar pressure measurement was then normalized by the total mean pressure to reduce the influence of walking speed on the magnitude—but not the distribution—of the plantar pressures. This normalization, proposed and validated by Keijsers [7], involves dividing each plantar pressure sample by the sum of all pixel values in the 2D mean pressure image $M$:

$$M(\mathbf{x}) = \frac{\sum_t V(\mathbf{x}, t)\ \delta[V(\mathbf{x}, t) > \tau]}{\sum_t \delta[V(\mathbf{x}, t) > \tau]}, \tag{1}$$

where $\tau$ = 5 kPa, $\delta$ is the Kronecker delta function, and the normalized plantar pressure measurement becomes $\tilde{V}(\mathbf{x}, t) = V(\mathbf{x}, t)/\sum_\mathbf{x} M(\mathbf{x})$. Finally, peak pressure images $I$ were computed from each plantar pressure measurement by retaining the maximum pressure values recorded at each pixel across the stance phase:

$$I(\mathbf{x}) = \max_t \tilde{V}(\mathbf{x}, t). \tag{2}$$

Additionally, the presence of multiple plantar pressure measurements from each foot allows us to align and average all peak pressure images from a foot in order to reduce biological and measurement noise [10]. To perform this task, we select at random one of the foot's peak pressure images as a reference, $I_{ref}$, then align all images $\{I_1, \cdots, I_K\}$ to the reference using a rigid spatial image registration (i.e. the rotation and translation of one image to match another). This rigid registration is computed by maximizing mutual information between the image pairs:

$$T_j = \text{argmax}_T\ MI(I_{ref}, I_j \circ T), \tag{3}$$

where $MI()$ is the histogram-based mutual information metric (with 50 bins per dimension), $T$ is a rigid transformation, and the $\circ$ operator represents the application of the transformation to the corresponding image. Mutual information is a probabilistic measure that encourages

homogeneous image regions to match to each other [40]. Eq 3 was optimized using the 1+1 evolutionary optimizer proposed by Styner et al. [41]. Once all peak pressure images from the same foot are aligned, we average the aligned images to produce a single representative image of the peak pressures:

$$\bar{I} = \frac{1}{K}\sum_{j=1}^{K} T_j \circ I_j,$$ (4)

where $K$ is the number of peak pressure images collected from the given foot.

Finally, we follow the convention of Pataky et al. by assuming that there is no common asymmetry pattern between a person's left and right plantar pressures, thereby allowing us to regard the plantar pressures from an individual's left and right feet as being essentially independent samples [42]. Using this independence assumption, we can group the left and right feet into a single statistical model. In this work, we chose to build a single "left" foot model. To do so, peak pressure images from right feet are flipped along the medial-lateral axis in order to simplify the subsequent model building and analysis steps.

## Statistical modelling

Fig 2 shows the workflow of the proposed statistical modelling in PAPPI. The workflow contains three main components. First, an anatomically-unbiased peak pressure template is created to which all measurements will be aligned for the subsequent analysis. Second, all peak pressure measurements are aligned to the template so that we have a pixel-by-pixel anatomical correspondence between all individuals' peak pressures. Finally, statistical models are built at each pixel to model both the relationship between the peak pressures and demographic factors as well as the fraction of the peak pressures that are not explained by the demographics. We introduce each of these steps below.

**Pairwise registration.** Pairwise registration is employed here to bring peak pressure images into alignment prior to building the statistical model. It is also used as part of the algorithm that generates the anatomically-neutral peak pressure template. This alignment step is required in order to establish anatomical correspondence between plantar pressures from different individuals, a correspondence that allows us to compute meaningful statistics pixel-by-pixel.

To perform this pairwise registration, we follow the registration framework proposed by Pataky et al. [42]. Given the peak pressure image $I_i$ of individual $i$, and a chosen template image $I_{template}$ (described in the following section), we first perform a rigid registration using Eq (3) to rigidly align $I_i$ to $I_{template}$, followed by a deformable registration using the peak pressure silhouette images. We define a peak pressure silhouette image, $S_i$, as

$$S_i(\mathbf{x}) = \begin{cases} 1 & \text{if } T_i \circ I_i(\mathbf{x}) > 5 \text{ kPa;} \\ 0 & \text{otherwise} \end{cases}$$ (5)

where $T_i$ is the rigid transformation obtained from Eq (3).

The silhouette image, $S_i$, is then non-rigidly aligned to the template's silhouette image, $S_{template}$, using diffeomorphic demons [43]. The diffeomorphic demons algorithm computes a deformation vector field $\phi$ that minimizes:

$$\begin{aligned} \phi = \text{argmin}_\phi & \quad \|S_{template} - S_i \circ \phi\|_2^2 + \lambda_1 \|\phi - \omega\|_2^2 + \lambda_2 \|\nabla\omega\|_2^2, \\ \text{subject to} & \quad \phi \in \mathcal{G} \end{aligned}$$ (6)

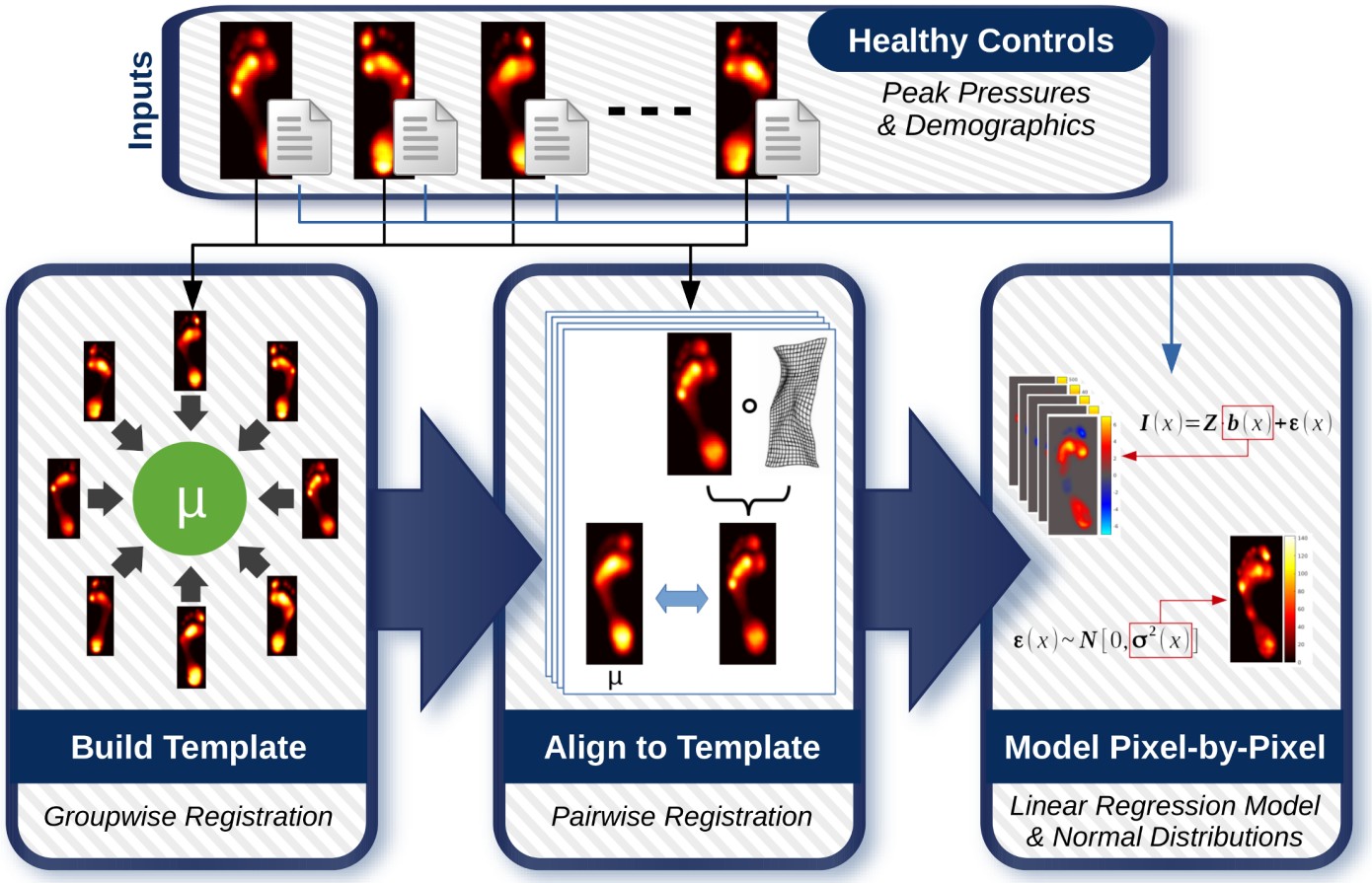

**Fig 2. Flow chart of statistical modelling workflow.** The proposed PAPPI technique begins by creating an anatomically-unbiased template to which all healthy peak pressure images are aligned. Once aligned, statistical models are built pixel-by-pixel in order to provide localized statistical analysis. Specifically, the images and their corresponding demographic factors are used to build (a) linear regression models of the peak pressures and (b) Normal distributions of the model residuals.

where $\mathcal{G}$ is the set of diffeomorphic transformations (i.e. invertible, smooth deformation vector fields) and the variable $\omega$ is an artificial variable introduced to split the optimization in two. One step of the optimization minimizes the first term that encourages similarity between the silhouette images, while the second step minimizes the third term to encourage the deformation to be smooth. Both optimization steps include the second term to encourage the smooth deformation, $\omega$, and the accurate deformation, $\phi$, to be equal. These optimization steps are interleaved until both converge. Finally, the regularization variables, $\lambda_1 = \lambda_2 = 2$, are empirically set to control the trade-off between the three objectives in the registration function. Once Eq (6) is optimized, the aligned peak pressure image is obtained as $I_i' = \phi_i \circ T_i \circ I_i$.

**Groupwise registration.** Generally speaking, statistical modelling and parametric mapping techniques like those used in PAPPI produce more accurate results when an anatomically-unbiased peak pressure image is chosen as a reference [42]. To create such an unbiased template image, we employed the groupwise registration algorithm of Guimond et al. [44] in combination with the pairwise registration technique introduced above. The groupwise registration algorithm consists of four main steps:

1. Randomly select an initial template image from the $N$ peak pressure images in the healthy controls dataset $I_{template} \in \{I_1, \cdots, I_N\}$,

2. Align $I_{template}$ to each peak pressure image $I_i \in \{I_1, \cdots, I_N\}$ using Eqs (3) and (6), thereby obtaining rigid transformations $T_i \in \{T_1, \cdots, T_N\}$ and deformation vector fields $\phi_i \in \{\phi_1, \cdots, \phi_N\}$,

3. Average the linear transformations and deformation vector fields from step 2 above to obtain $\bar{T}$ and $\bar{\phi}$. These transformations capture how $I_{template}$ differs from the population average,

4. Apply the average transformations to the template image $I_{template} = \bar{\phi} \circ \bar{T} \circ I_{ref}$. By applying these transformations, we move $I_{template}$ towards the average foot shape and size.

These four steps are repeated until no further change in $I_{template}$ is seen.

Once this algorithm has been run and an unbiased template image $I_{template}$ has been obtained, all peak pressure images are aligned to this template using the pairwise image registration algorithm described in the previous section.

**Statistical model building.** Once all peak pressure images are anatomically aligned, we model—pixel-by-pixel—the relationship between the peak pressures and demographic factors. In this work, we use the demographic factors of age, sex, weight, height, and shoe size, resulting in the demographics vector $\mathbf{z}_i = [age_i, sex_i, weight_i, height_i, shoe\ size_i, 1]$ for the individual $i$. At each pixel, $\mathbf{x}$, we employ ordinary least squares to fit a linear regression model

$$\mathbf{y}(\mathbf{x}) = \mathbf{Z}\mathbf{b}(\mathbf{x}) + \epsilon(\mathbf{x}), \tag{7}$$

where $\mathbf{y}(\mathbf{x}) = [I'_1(\mathbf{x}), \cdots, I'_N(\mathbf{x})]^T$ are the peak pressures at pixel $\mathbf{x}$, $Z = [\mathbf{z}_1, \cdots, \mathbf{z}_N]^T$ is a matrix containing the corresponding participant's demographic factors, and $\mathbf{b}(\mathbf{x})$ are the model parameters that indicate the influence of each factor on the peak pressures. The residuals, $\epsilon(\mathbf{x})$, are subsequently modelled as being uncorrelated and following a zero-centred normal distribution

$$\epsilon(\mathbf{x}) \sim \mathcal{N}(0, s^2(\mathbf{x})), \tag{8}$$

where $s^2(\mathbf{x})$ is the sampled variance of the residuals at pixel $\mathbf{x}$.

It is important to note that two items are being modelled here: the effect of demographic factors is modelled in Eq (7), and the remaining peak pressure variances are statistically modelled in Eq (8) through normal distributions on the residuals. Additionally, these statistical models were built only for the pixels within the silhouette of the reference image $S_{template}$. This constraint is applied as the registration steps should reduce the presence of non-zero peak pressures outside this region and, therefore, areas outside this region are unlikely to have peak pressures that satisfy the normality assumption expressed in Eq (8).

## Statistical testing

The personalized evaluation of a patient's peak pressure image follows the workflow shown in Fig 3 and is similar to outlier detection techniques seen elsewhere [31, 32]. The patient's demographic factors (age, sex, weight, height, shoe size) are inputted into our linear regression model to predict their peak pressure image. The patient's real peak pressures, acquired by measurement, are then aligned to the predicted image in order to obtain a pixel-by-pixel anatomical correspondence between the two peak pressure images. Once aligned, the residuals between the measured and predicted peak pressures are computed and compared to the

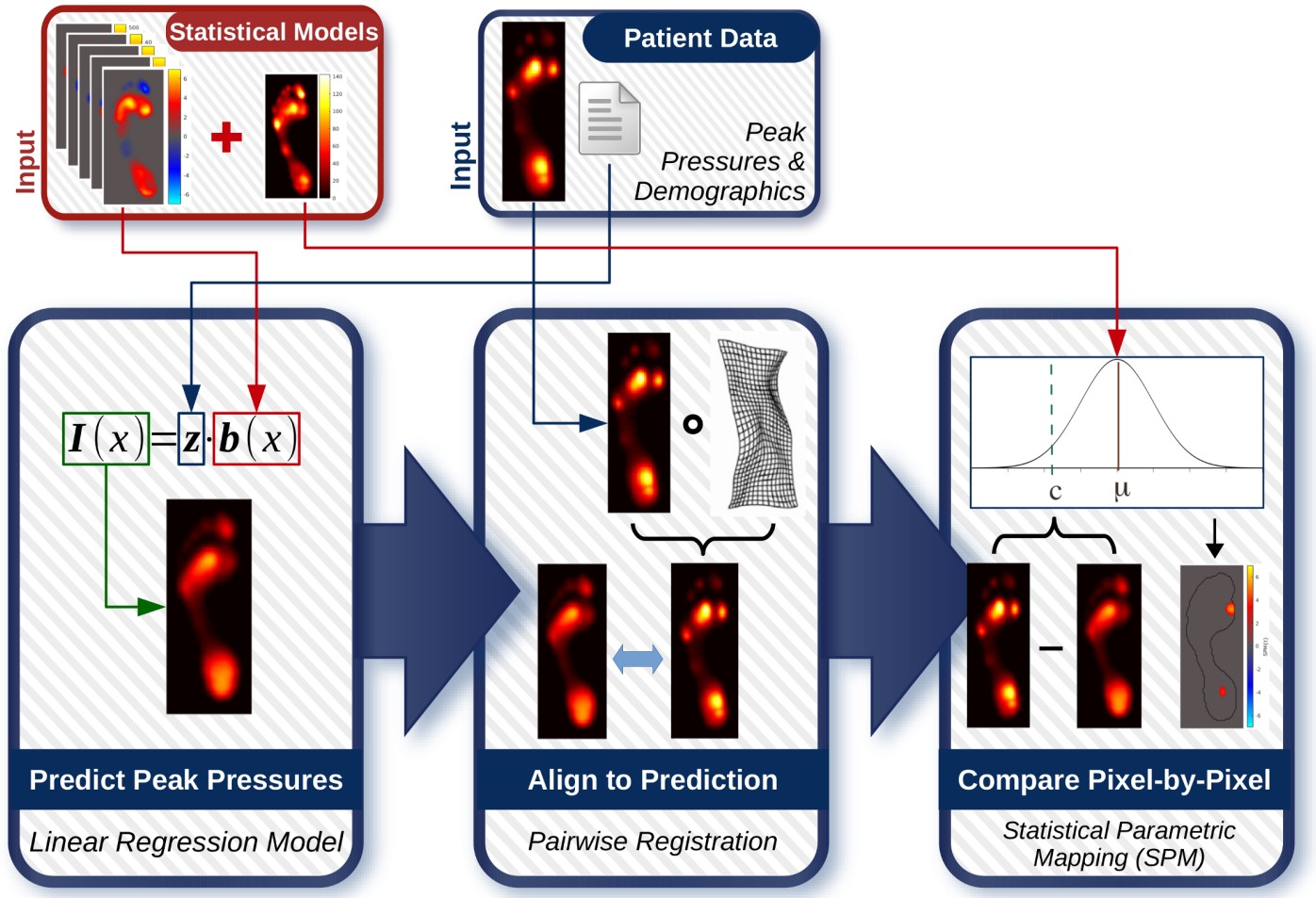

**Fig 3. Flow chart of statistical testing workflow.** Given a peak pressure image and demographic characteristics from a new patient, a healthy peak pressure image for the patient is estimated using the statistical model. The patient's measured pressures are then aligned to the estimated image, and a residual image is created by subtracting the estimated pressures from the measured ones. Finally, statistical parametric mapping is used with single-sample t-tests in order to identify patient residuals that are outliers from the statistical model's Normal distributions over the residuals.

Normal distributions in our statistical model. Any outlier pressures are then highlighted using SPM. Details on these steps are outlined below.

**Predict pressures.** Given a new patient's demographic information in the vector $\mathbf{z}_{new}$ = [$age_{new}$, $sex_{new}$, $weight_{new}$, $height_{new}$, shoe $size_{new}$, 1], their peak pressure image is estimated from our linear regression model as

$$I_{predict}(\mathbf{x}) = \mathbf{z}_{new}\mathbf{b}(\mathbf{x}), \qquad (9)$$

where $\mathbf{b}(\mathbf{x})$ are the linear regression coefficients estimated at pixel $\mathbf{x}$ using Eq (7).

**Pairwise registration.** To analyze the new patient's peak pressures pixel-by-pixel, we first align their peak pressure image $I_{new}$ to their predicted peak pressure image $I_{predict}$ from Eq (9). This alignment is performed using the pairwise image registration framework described in Eqs (3) and (6). The aligned image $I'_{new} = \phi_{new} \circ T_{new} \circ I_{new}$ is then used for further analysis.

**Statistical parametric mapping.** Once the patient's peak pressure image is aligned to its prediction, we compute the residuals $R$ as

$$R(\mathbf{x}) = I'_{new}(\mathbf{x}) - I_{predict}(\mathbf{x}). \qquad (10)$$

Based on our statistical model, we assumed that these residuals are sampled from the Normal distributions defined in Eq (8). To test whether this assumption is valid for a new patient, we compute single-sample t-tests on the residuals at each pixel to create a statistical parametric map of t-statistics:

$$SPM\{t\}(\mathbf{x}) = \frac{R(\mathbf{x})}{\epsilon(\mathbf{x})/\sqrt{N}}. \qquad (11)$$

Note that the use of single-sample t-tests is equivalent to the DisCo-Z approach that is seen in other subject-specific abnormality studies [45]. Random field theory was then used on the resulting statistical parametric map, $SPM\{t\}$, to identify pixels, and clusters of pixels, whose peak pressures are significantly different (at $\alpha = 0.05$) than those predicted by the statistical model [46]. T-statistics that are below the significance threshold are zeroed out and the resulting statistical outliers in $SPM\{t\}$ are highlighted for display.

## Results

Fig 4 shows an example result of the PAPPI algorithm for a hallux valgus patient. The patient's peak pressure image is shown in Fig 4(a) while the model's prediction of their baseline peak pressures is shown in Fig 4(c). PAPPI aligns the measured pressures to the predicted pressures, the result of which is shown in Fig 4(b). It is worthwhile at this point to evaluate the quality of the alignment and we do so by superimposing the aligned image (in blue) to the predicted image (in red) in Fig 4(d). In this example, as in all images examined in our study, any alignment error appeared to be minimal. The raw single-sample t-statistics for this patient's peak pressures are shown in Fig 4(e), and after thresholding these t-statistics for significance, we obtain the abnormality map shown in Fig 4(f). Significantly higher peak pressures were observed under metatarsal 1 compared to our modelled healthy controls. It is these pressures that PAPPI highlights for a clinician's further analysis. Similar figures for all 69 hallux valgus cases in our study are shown in S1 File.

To summarize all of our PAPPI results, we present them according to which foot regions contained abnormal peak pressures: the heel, midfoot, metatarsal 1, metatarsals 2-5, and toes. We defined these regions using the Novel 10 region mask [7], with some regions merged (e.g. medial and lateral heel, metatarsals 2-5, hallux and other toes) in order to increase the statistical power in subsequent following experiments. Note that, in contrast to Table 1, we combined the hallux and lesser toes into a single group, as well as isolating metatarsal 1 from the rest of the forefoot, as these groupings aligned more closely with the results we observed. Some patients presented with abnormal pressures in more than one region while others presented with no abnormal plantar pressures.

The abnormal pressure pattern most frequently seen in our hallux valgus cohort is the one shown in the example highlighted above: abnormally high peak pressures under metatarsal 1. Out of our 69 hallux valgus cases, 26 of them showed this pattern (38%). All 26 of these cases are displayed in Fig 5.

The second-most frequently seen abnormal pressure pattern is seen in the toes. Of the 69 hallux valgus cases in our study, 25 of them showed increased pressures under toes 2-5 and, occasionally, decreased pressures under the hallux (36%). All 25 of these hallux valgus cases

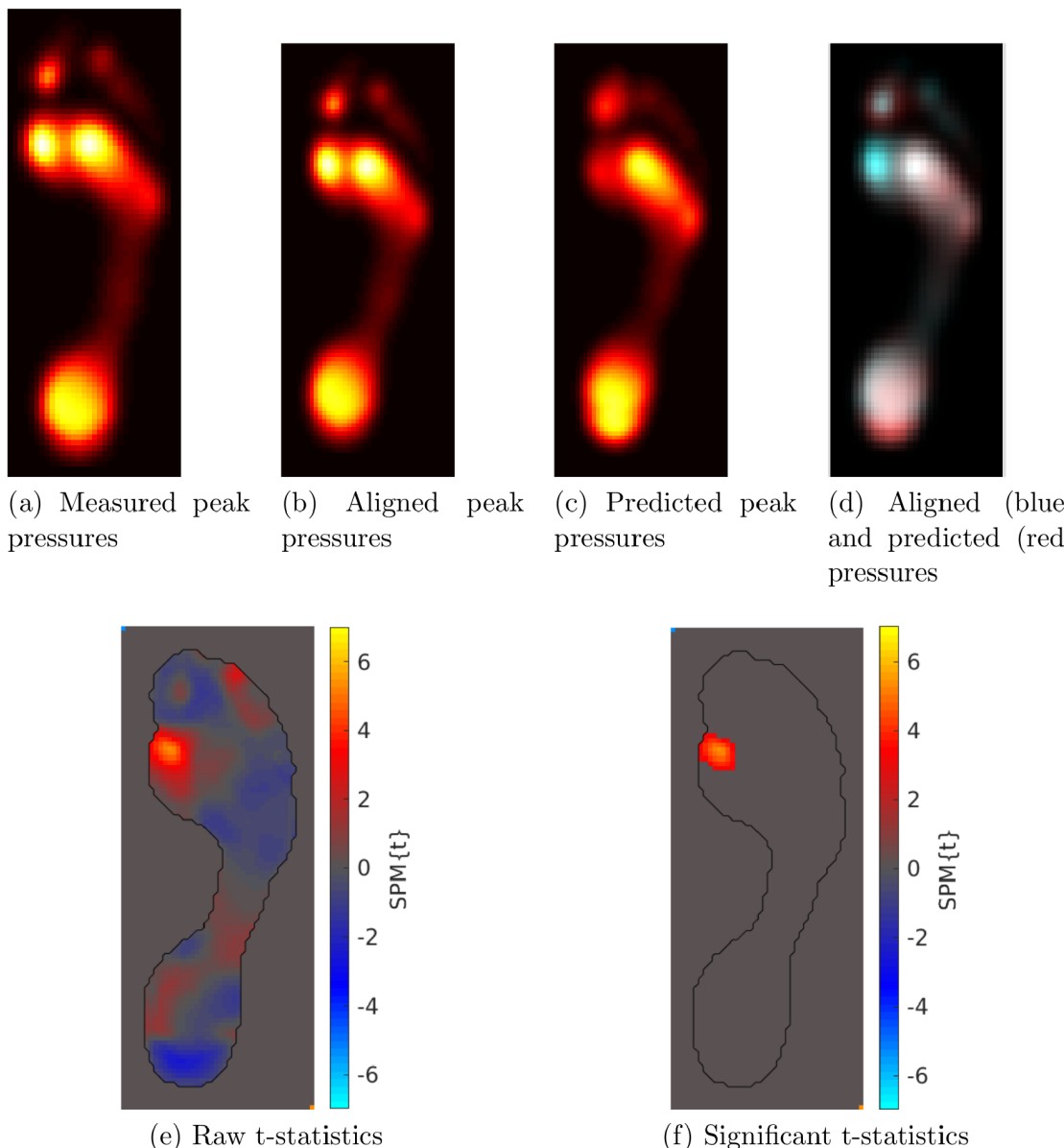

**(a)** Measured peak pressures

**(b)** Aligned peak pressures

**(c)** Predicted peak pressures

**(d)** Aligned (blue) and predicted (red) pressures

**(e)** Raw t-statistics

**(f)** Significant t-statistics

**Fig 4. Example of PAPPI Output.** Given a patient's peak pressure image (a), it is aligned (b) to the peak pressure image predicted for this patient by the statistical model (c). The aligned (blue) and predicted (red) images are superimposed (d) to ensure that an accurate alignment between them has been achieved. Once aligned, single-sample t-statistics are computed at each pixel (e) and random field theory is used to test for significance (f).

are shown in Fig 6. These abnormalities may indicate a more lateral roll off of the toes in these hallux valgus patients.

The next-most observed abnormal pressure pattern includes increased peak pressures in the midfoot. This pattern of abnormal pressures is commonly seen in individuals with pes planus (i.e. a flat foot arch) [47, 48]. Of the 69 hallux valgus cases in our study, 24 of them showed this pes planus pressure pattern (35%). These 24 cases are shown in Fig 7.

A fourth group of patients displayed abnormal peak pressures in the forefoot and outside metatarsal 1. Of the 69 hallux valgus cases studied, 16 presented abnormal peak pressures in this area (23%). These 16 cases are presented in Fig 8.

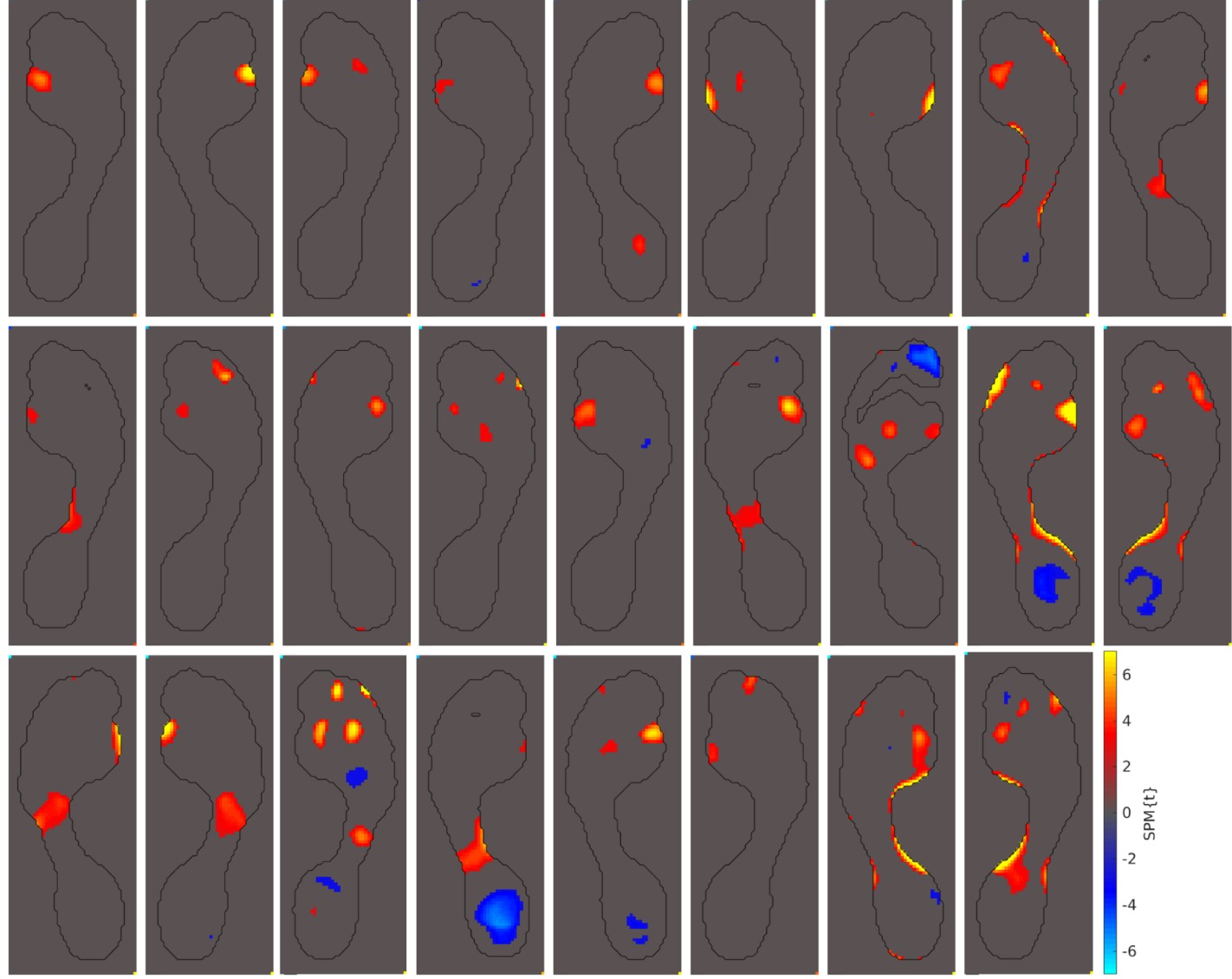

**Fig 5. Patients with increased MT 1 pressures.** Out of the 69 hallux valgus cases we examined, these 26 displayed abnormally high peak pressures under metatarsal 1 (38%).

A fifth pattern of abnormal peak pressures is also occasionally seen: abnormal peak pressures under the heel. Of the 69 hallux valgus cases studied, 13 of them show this abnormality pattern (19%). They are shown in Fig 9. In addition to these 5 abnormal peak pressure patterns, 16 of the 69 hallux valgus cases studied showed no peak pressure abnormalities at all (23%).

To get a further understanding of these abnormal peak pressure patterns, we compared the presence of each pattern with the foot function scores obtained from the FFI-5pt and MOXFQ questionnaires. For each abnormality pattern, unpaired t-tests were performed on the foot function scores between cases with the abnormality pattern and cases without. Over all five abnormality patterns, only the MOXFQ showed significant differences, and only for these groups shown in Fig 10. Hallux valgus patients with abnormally high peak pressures under metatarsal 1 showed significantly lower MOXFQ scores ($p = 0.011$) than those patients who

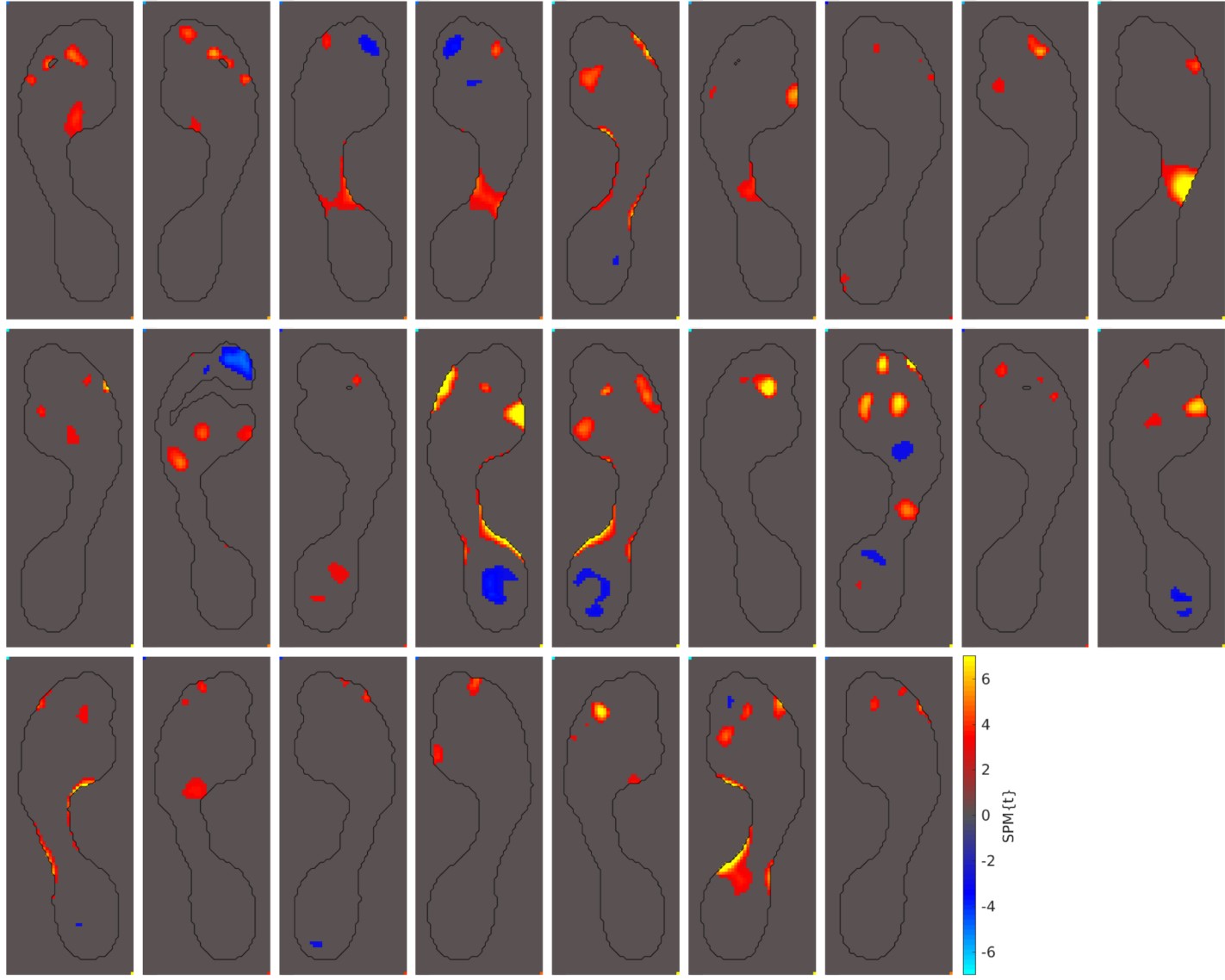

**Fig 6. Patients with abnormal pressures under the toes.** Out of the 69 hallux valgus cases we examined, these 25 displayed abnormally high peak pressures under toes 2-5 and, occasionally, abnormally low pressures under the hallux (36%).

did not show this abnormal pressure pattern. Conversely, patients with abnormal heel pressures showed higher foot pain scores on the MOXFQ than those who did not ($p = 0.014$). However, after correcting for multiple comparisons using false discovery rate [49], both results lose statistical significance (corrected $\alpha = 0.002$). No significant differences in FFI-5pt scores were seen.

Similarly, we compared both the radiographic IMA and HVA measurements to the observed abnormality patterns. Once again, unpaired t-tests were used to compare the angles between patients with the abnormality pattern and those who do not. Fig 11 show the patterns that contained significant HVA differences. Hallux valgus patients with abnormal heel pressures showed significantly higher HVA than those who did not. Conversely, patients with no plantar pressure abnormalities showed significantly lower HVA than those who did. Again,

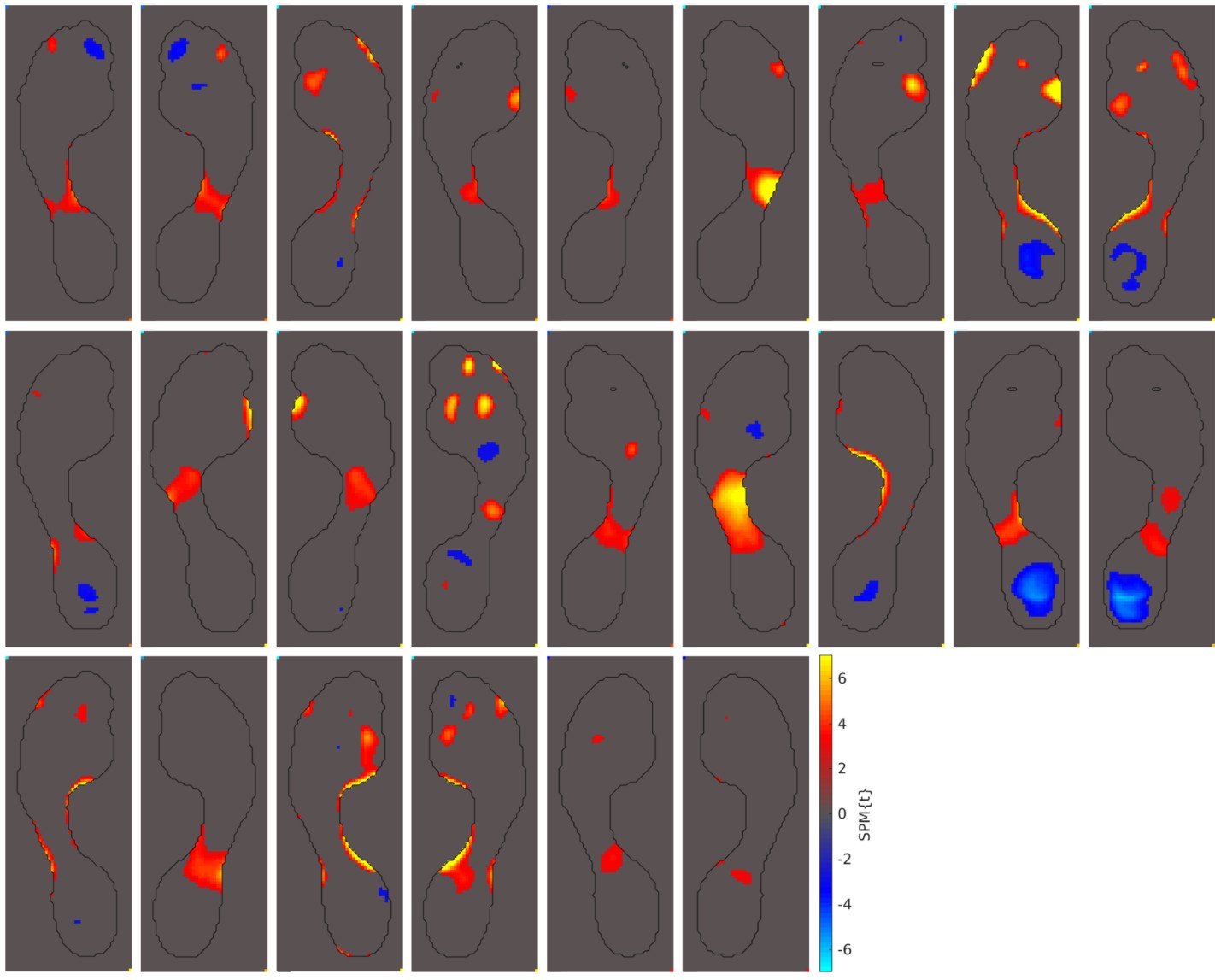

**Fig 7. Patients with pes planus pressure patterns.** Out of the 69 hallux valgus cases we examined, these 24 displayed abnormally high peak pressures under the midfoot (35%). These abnormality patterns have previously been seen in individuals with pes planus [47, 48].

these results lose statistical significance after correcting for multiple comparisons using false discovery rate. No significant differences in IMA were seen.

Finally, we evaluated the normality assumption on the residuals in our statistical model using Kolmogorov-Smirnov tests [50]. Multiple comparison correction was performed using false discovery rate. These results are shown in Fig 12. Of the 2024 pixels within our model, the null hypothesis (that our residuals are sampled from the normal distributions used in our model) was rejected in only 57 of them (2.8%, corrected $\alpha$ = 0.0014). Most pixels that reject this hypothesis appear around the edges of the heel and the toes. The lone exception is a 4-pixel large area near metatarsal 5 where the peak pressure distributions are skewed towards higher pressures. Note that none of our results reported abnormally high pressures in this 4-pixel region, suggesting that this skew in the modelled residuals did not impact our analysis.

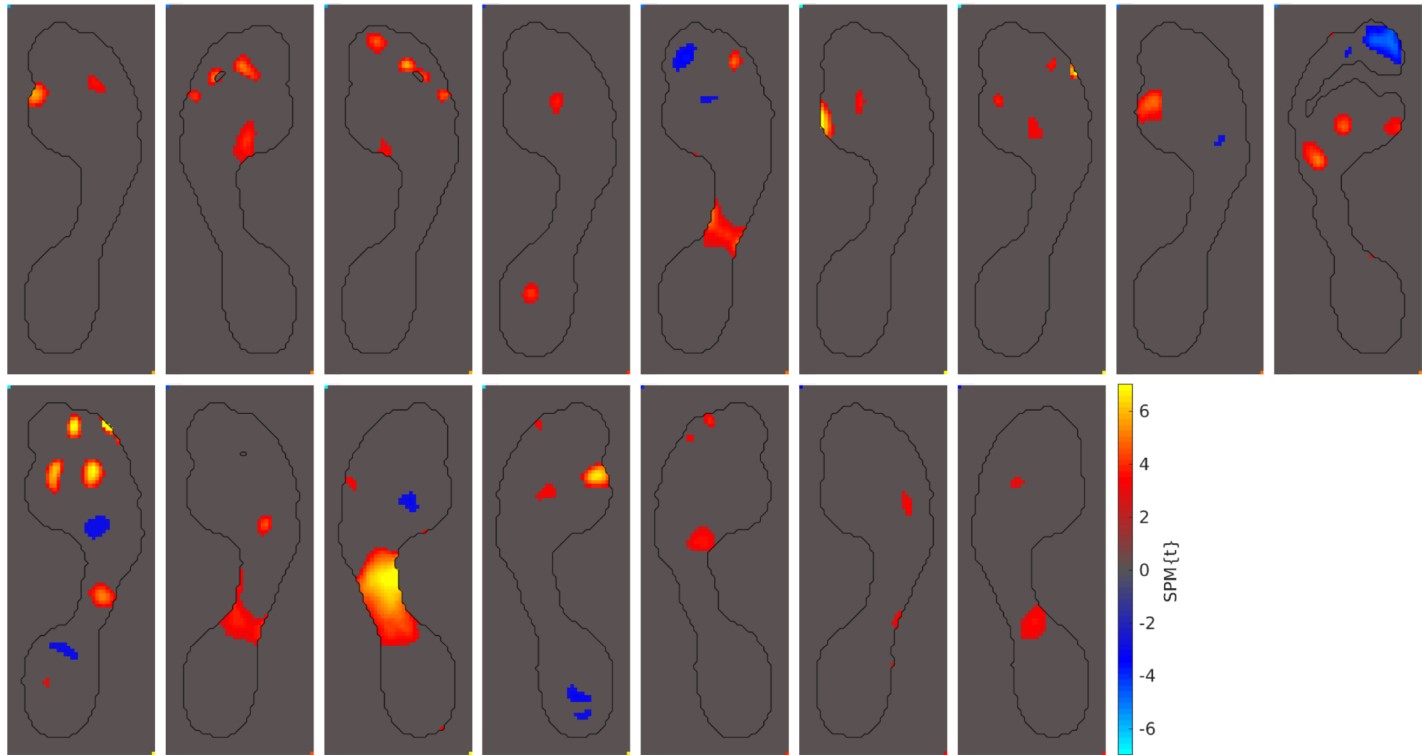

**Fig 8. Patients with abnormal pressures under MT 2-5.** Out of the 69 hallux valgus cases we examined, these 16 displayed abnormal peak pressures under metatarsals 2-5 (23%).

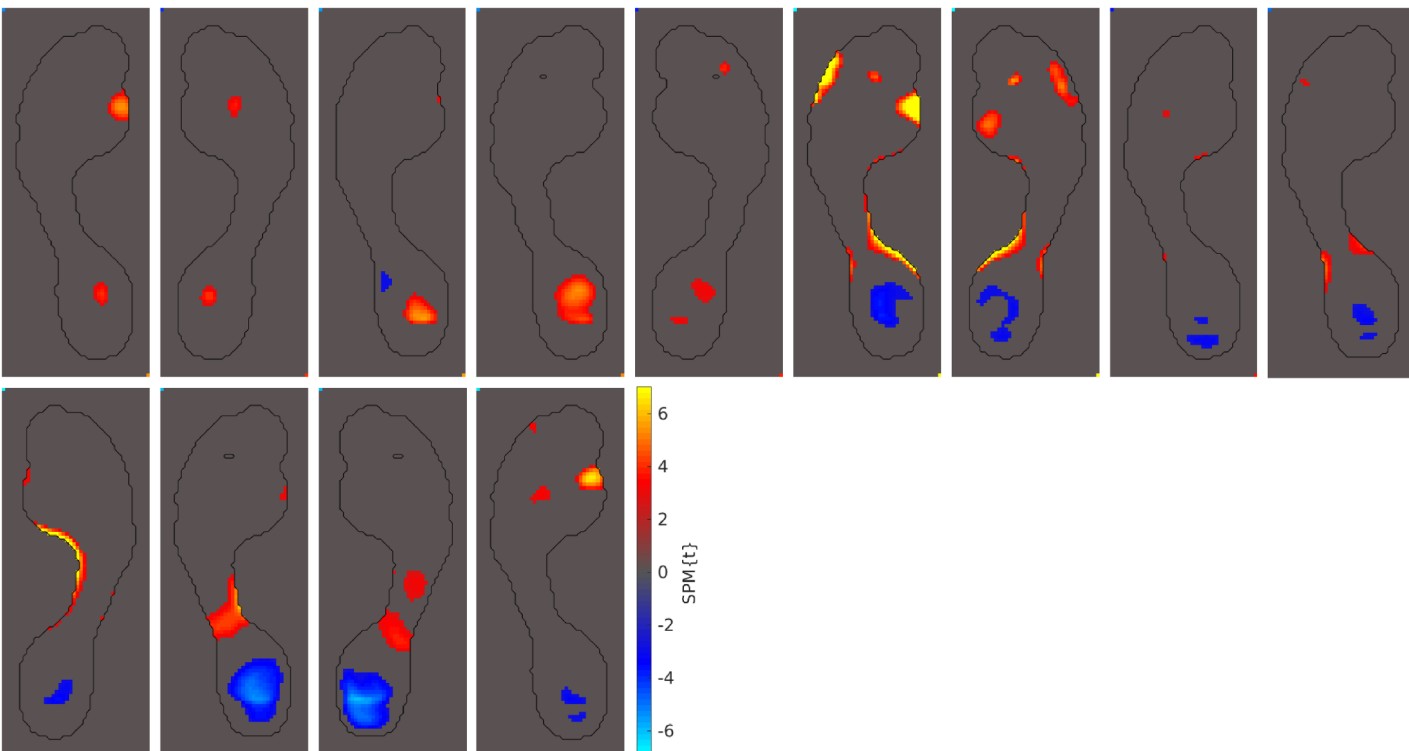

**Fig 9. Patients with abnormal pressures under heel.** Out of the 69 hallux valgus cases we examined, these 13 displayed abnormal peak pressures under the heel (19%).

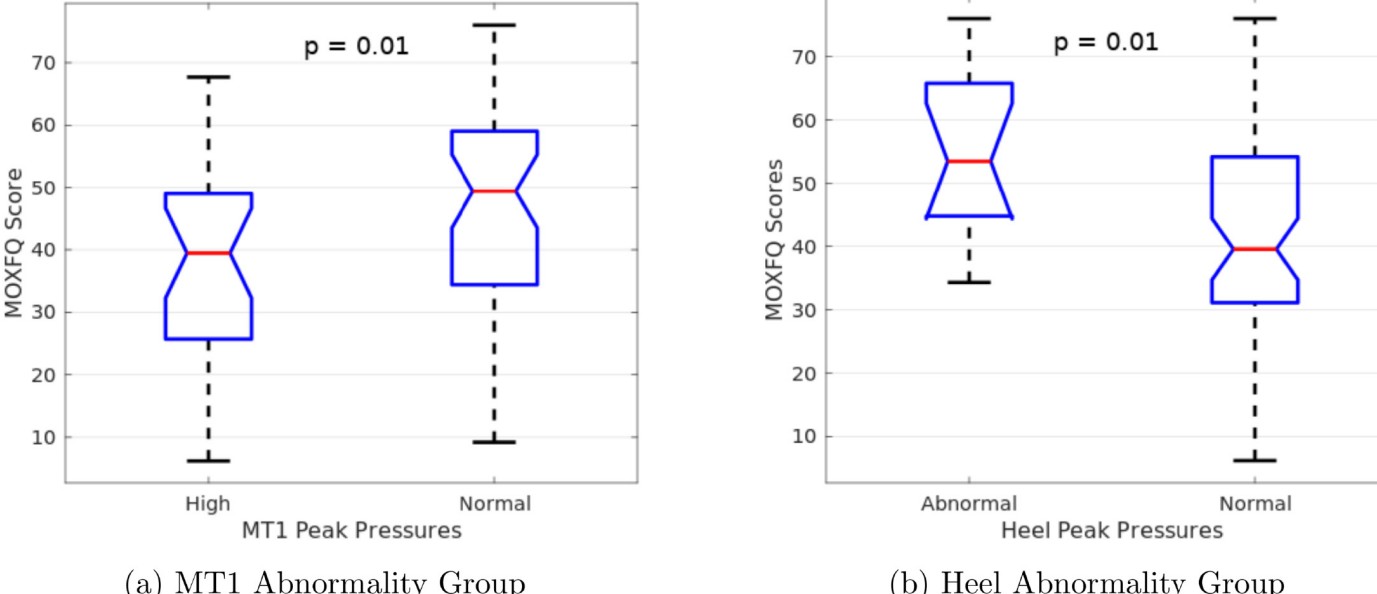

(a) MT1 Abnormality Group (b) Heel Abnormality Group

**Fig 10. Significant t-test results for Manchester-Oxford Foot Questionnaire Scores (MOXFQ).** Hallux valgus patients with abnormally high pressures under metatarsal 1 showed lower foot pain scores on the MOXFQ than those who did not ($p = 0.011$). Conversely, patients with abnormal heel pressures showed higher foot pain scores on the MOXFQ than those who did not ($p = 0.014$). After performing a false discovery rate correction, both results lose statistical significance ($\alpha = 0.002$).

## Discussion

When examining the abnormal pressure patterns identified by PAPPI, we noted that they share similarities with results from previous group studies on hallux valgus patients. For

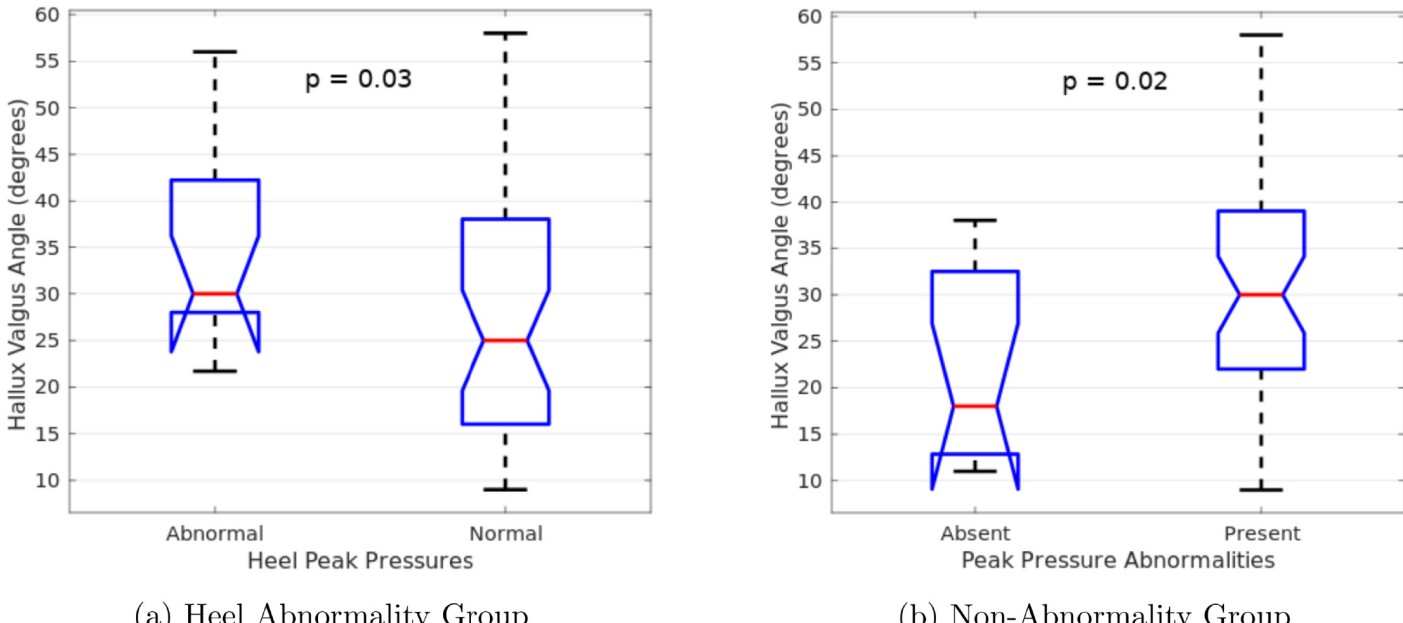

(a) Heel Abnormality Group (b) Non-Abnormality Group

**Fig 11. Significant t-test results for hallux valgus angles.** Hallux valgus patients with abnormal heel pressures showed higher hallux valgus angles than those who did not ($p = 0.033$). Conversely, patients that showed no pressure abnormalities had lower hallux valgus angles than those who did ($p = 0.018$). After performing a false discovery rate correction, both results lose statistical significance ($\alpha = 0.002$).

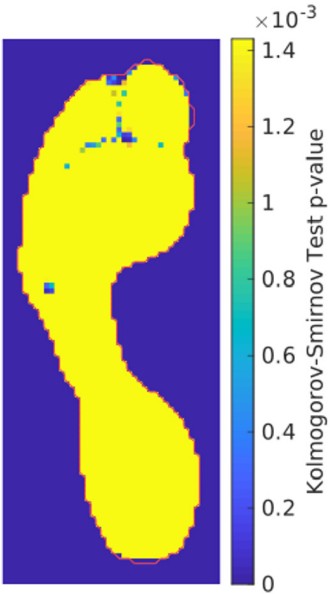

**Fig 12. Normality of residuals in statistical model.** P-values from Kolmogorov-Smirnov tests that evaluate the goodness of fit for the normal distributions over the residuals in our statistical model. Note that only 2.8% of the pixels in the model reject this hypothesis (corrected $\alpha = 0.0014$). The image region within the statistical model is outlined in red.

example, the increases in peak pressure under metatarsal 1 were also found in the studies of Bryant et al. and Wen et al. [13, 16]. Similarly, the pes planus abnormality pattern, with its increases in midfoot pressures, was also reported previously [15]. Decreases in hallux pressures and increases in pressures under toes 2-5 would also agree with the work of Galica et al. [12], while pressure abnormalities seen under metatarsals 2-5 may relate to similar pressure differences reported in previous studies [12–16]. We hypothesize that the discrepancies in previously-reported results may simply come down to how many patients in those studies present with one or more of our identified pressure abnormality patterns. It may also depend on how many of the selected patients showed no abnormal peak pressures at all, as 23% of our cohort did. Ultimately, the variability in the results produced by PAPPI may disambiguate the results seen at the group level in previous studies, thereby highlighting the need and the value of this personalized analysis technique in evaluating plantar pressures.

Additionally, the abnormal pressure patterns identified by PAPPI suggest some intuitive interpretations. The pressure abnormalities in the toes may be explained by a more lateral roll-off from the toes for hallux valgus patients compared to healthy controls. The fact that 36% of our hallux valgus patients also showed evidence of pes planus, indicated by increased pressure under the midfoot [51], suggests that individuals with flat feet may be more susceptible to developing a hallux valgus than the rest of the population (note that body weight was included in our modelling, so arch height is more likely to explain these results). The results showing lower MOXFQ scores for patients with higher pressures under metatarsal 1 could indicate that the pain these patients experience is not yet strong enough for them to begin unloading metatarsal 1. The lower hallux valgus angles for patients that show no plantar pressure abnormalities also suggest that mild hallux valgus cases do not significantly alter one's gait. Finally, abnormal pressures in the heel suggest more painful and severe hallux valgus cases, cases that may require a patient to significantly alter their gait. These interpretations of PAPPI's results

could be used as hypotheses in future studies, thereby also highlighting PAPPI's value as a personalized plantar pressure exploration and research tool.

While the current study applies PAPPI to peak pressure images for the evaluation of hallux valgus patients, there are few limitations to the technique. Mean pressure and pressure-time integral images could just as equally be used in PAPPI, and other patient groups could be examined so long as the image registration steps result in accurate footprint alignments. Additionally, a variety of demographic factors could be included in the statistical modelling (e.g. footedness, leg length, foot progression angle). PAPPI could also be used in region-of-interest studies or centre of pressure studies by simply removing the registration steps from the work flow. Similarly, PAPPI could be extended to plantar pressure videos by incorporating the dynamic time warping used in STAPP [11]. The general applicability of PAPPI and its extensions to other plantar pressure measures are items we intend to examine in future work. We have also shared the MATLAB code for PAPPI as supplementary material (S2 File) in order to allow other groups to use and extend PAPPI as they see fit.

Despite the strengths of PAPPI, we would caution against over-interpretation of the results presented here. One key reason for exercising caution is the limited number of individuals (55) included in the statistical model. When building statistical models, it is typically advised that the model satisfy the one-in-ten rule: that one predictive variable is studied using a minimum of ten events [52]. In this work, our pixel-by-pixel statistical models contain 7 predictive variables: the five linear regression coefficients for the demographics factors, the bias variable in the linear regression, and the standard deviation of the residuals. Using the one-in-ten rule, at least 70 individuals would be desirable for building the statistical model and ensuring reasonable validity. We are currently unable to hit this threshold due to a lack of available data. As a result, it may be the case that some of the abnormalities observed in the hallux valgus patients may be due to the fact that the 55 individuals in our statistical model provide an incomplete view of a true healthy population. There is some evidence that this might be the case since the abnormalities seen in some foot regions did not correlate with either the foot function scores or the radiographic measurements. Nevertheless, what we have shown is a proof-of-concept for the methodology behind PAPPI. A full validation of PAPPI, including an evaluation of the one-in-ten rule for this framework, is planned as future work.

It should also be noted that PAPPI makes two simplifying assumptions in the statistical modelling. First, we assume that the relationship between peak pressures and demographic factors is a linear one. While there is some evidence to support this claim [18, 19], it does not rule out the possibility that a non-linear regression model could improve upon these results. Consequently, the use of non-linear models is something we are currently exploring in our CAD WALK project (http://cadwalk.eu). Second, we assume that the residuals in our linear regression model follow a normal distribution. While this was generally the case in our study (see Fig 12), the validity of this assumption is not always guaranteed. We would recommend that this assumption be checked whenever PAPPI is used. This normal distribution assumption will also affect the amount of the midfoot that gets evaluated. PAPPI currently uses the plantar surface area of the template as a boundary in order to increase the chances that this normal distribution assumption holds for all statistical analyses. Unfortunately, this boundary can omit some of the plantar surface area measured from individuals with pes planus. Nevertheless, PAPPI was able to observe the effects of pes planus through increased midfoot pressures (see Fig 7). Removing this normal distribution assumption, thereby allowing for modelling of variable surface areas, is also a potential area of future work.

Finally, there are situations where the use of PAPPI may not be advised. If a patient has a severe foot deformity or pathological condition, the registration algorithms used by PAPPI may not be able to bring their pressure measurement into alignment with a prediction from

our statistical model. In this situation, the statistical testing would not compare plantar pressures from similar locations on the foot, resulting in statistics that are not meaningful. To validate PAPPI in our study, we qualitatively confirmed the accuracy of image registrations using visualizations like those in Fig 4(d) and in S1 File. We recommend that these checks be performed whenever PAPPI is used in order to make sure that the abnormal pressures highlighted by PAPPI are indeed meaningful statistical outliers. Given successful registration results, validated normality assumptions, and enough plantar pressure measurements, PAPPI can combine the localized analysis of SPM techniques with the ability to analyze individual plantar pressure measurements, a combination that has yet to appear in the plantar pressure analysis literature.

## Conclusion

We have introduced PAPPI as a statistical framework for the personalized analysis of plantar pressure images. PAPPI statistically models plantar pressures and demographics of healthy controls so that a personalized baseline can be created for an individual's plantar pressures. These baseline estimates, combined with Normal distributions on the errors of the estimates, are then compared to the individual's real plantar pressures using single-sample t-tests. The resulting t-statistic maps, thresholded for statistical significance, help highlight where an individual's plantar pressures are abnormal. When applying PAPPI to a cohort of hallux valgus patients, we observed 5 abnormality patterns, patterns which overlap with those observed in previous group-level studies. Unlike those previous group studies, PAPPI is able to point out that hallux valgus patients have rather heterogeneous plantar pressures and suggests that this heterogeneity may have an impact on both a person's susceptibility to this foot deformity as well as the amount of foot pain they experience. While further validation of PAPPI is necessary, its ability to provide an intuitive, quantitative, and personalized plantar pressure analysis makes it unique as a plantar pressure exploration and research tool.

## Supporting information

**S1 File. All hallux valgus results.** Plantar pressure predictions, image registration results, and statistical parametric maps (thresholded and non-thresholded) for each of the 69 hallux valgus cases in this study.
(PDF)

**S2 File. MATLAB code for PAPPI.** The source code for PAPPI is provided in MATLAB format along with a readme file describing how it can be used.
(ZIP)

## Author Contributions

**Conceptualization:** Brian G. Booth, Toon Huysmans.

**Data curation:** Brian G. Booth, Eva Hoefnagels, Noël L. W. Keijsers.

**Funding acquisition:** Brian G. Booth, Jan Sijbers.

**Investigation:** Brian G. Booth, Noël L. W. Keijsers.

**Methodology:** Brian G. Booth.

**Project administration:** Brian G. Booth, Toon Huysmans, Jan Sijbers.

**Resources:** Noël L. W. Keijsers.

**Software:** Brian G. Booth.

**Supervision:** Jan Sijbers, Noël L. W. Keijsers.

**Validation:** Brian G. Booth, Eva Hoefnagels, Noël L. W. Keijsers.

**Visualization:** Brian G. Booth.

**Writing – original draft:** Brian G. Booth.

**Writing – review & editing:** Eva Hoefnagels, Toon Huysmans, Jan Sijbers, Noël L. W. Keijsers.

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
