## [Decision Letter · Decision Letter 0]

13 Dec 2019

PONE-D-19-27376

PAPPI: Personalized analysis of plantar pressure images using statistical modelling and parametric mapping

PLOS ONE

Dear Dr. Booth,

Thank you for submitting your manuscript to PLOS ONE. After careful consideration, we and the reviewers feel that it has considerable merit but does not fully meet PLOS ONE’s publication criteria as it currently stands. Specifically, the reviewers have provided a number of mostly minor comments for addressing. Therefore, we invite you to submit a revised version of the manuscript that addresses the points raised during the review process.

Also, the zenodo data link does not specifically to a data file, but to a search page of different files. Please link directly to the complete dataset, ideally using a DOI. Please also be sure to correct for any other statistical analyses performed on the open dataset in other publications, or talk about such multiple comparisons issues briefly ...

We would appreciate receiving your revised manuscript by Jan 27 2020 11:59PM. To enhance the reproducibility of your results, we recommend that if applicable you deposit your laboratory protocols in protocols.io, where a protocol can be assigned its own identifier (DOI) such that it can be cited independently in the future. For instructions see: http://journals.plos.org/plosone/s/submission-guidelines#loc-laboratory-protocols

We look forward to receiving your revised manuscript.

Kind regards,

Manoj Srinivasan, Ph.D.

Academic Editor

PLOS ONE

Journal Requirements:

Reviewers' comments:

Reviewer's Responses to Questions

**Comments to the Author**

1. Is the manuscript technically sound, and do the data support the conclusions?

Reviewer #1: Yes

Reviewer #2: Yes

2. Has the statistical analysis been performed appropriately and rigorously? 

Reviewer #1: Yes

Reviewer #2: Yes

3. Have the authors made all data underlying the findings in their manuscript fully available?

Reviewer #1: Yes

Reviewer #2: Yes

4. Is the manuscript presented in an intelligible fashion and written in standard English?

Reviewer #1: Yes

Reviewer #2: Yes

5. Review Comments to the Author

Reviewer #1: Manuscript Review of PONE-D-19-27376: PAPPI: Personalized analysis of plantar pressure images using statistical modelling and parametric mapping

GENERAL COMMENTS

The paper presents a very innovative approach to assessing plantar pressure patterns on an individual level in order to better assess pathologies such as hallux valgus.

The paper is generally well written and represents an excellent contribution to the field. A few sections and the flowchart figures could be clearer. In the current form, it would be difficult for another researcher to reproduce this mythology in their lab, or for someone who is not very well versed in plantar pressure analyses to clearly understand the paper. Specific sections needing more clarity are outlined below in the specific comments.

One other general comment relates to describing the accuracy of the statistical model in more detail. You describe PAPPI as, “The idea behind PAPPI is to statistically model, in a layered way, the healthy population plantar pressures and the demographic factors that can influence them.” PAPPI is explained as being built off of 55 healthy controls. The model seems to then be directly used to test hallux valgus patients as a proof of concept of how the model can be used to identify/assess pathology. There is potentially a step missing that would involve showing that the model accurately predicts a healthy control. Was the model performance validated in any way? Since the model uses a 95% confidence interval, we would expect 95% of healthy subjects tested to show no statistical differences from their estimated pressure to their actual pressure. The paper references 23% of hallux valgus patients showing no difference from their estimated “healthy” plantar pressure. “Note that this number is much lower than the 95% of healthy cases we would expect to be free of abnormalities based on the definition of our statistical model.” Was this statement - the accuracy of the model - validated? Without information on the accuracy of the estimated plantar pressure there’s the potential to draw some incorrect conclusions based on it. Maybe this is another paper, and not necessarily needed for the current paper, but I would guide caution in interpreting results too much until the accuracy is verified.

SPECIFIC COMMENTS

P4, L100. Any concern with the healthy controls and hallux valgus patients being collected at different sampling frequencies? (500Hz vs. 200 Hz)

P5. Figure 1. The flow chart is not clear to me. What is meant by “a linear model of the peak pressures at each pixel”, and “Normal distributions of the model residuals”? I have a similar issue for Figure 2 – would be great to be able to understand a big picture of what you did from looking at the diagram.

P5. L146. This paragraph is not clear. You mention that you consider the left and right foot as being independent, but then in the last sentence of the paragraph (line 150) you mention that you flip the right foot to treat each image as containing peak pressures from the left? Please re-write in a clearer way.

P9. L276. Can you describe how each region of the foot was defined?

P11. L337. One aspect of the discussion that is missing is how the surface area that makes contact with the ground of each individual is considered. In the figures, it appears as though all plantar pressure patterns have a similar surface area (seems like a high-arch individual). Can the surface areas that make contact with the pressure pad be used in the assessment? For example, flat-footed individuals would have more surface area at the medial midfoot.

P11. L351. Typo: remove word “in”.

Reviewer #2: In your manuscript, you introduce a general statistical method for quantitative analyze of the image. GLMs are robust tools that can handle high dimension linear relation and should be of interest for the bio imaging field. In this manuscript, authors propose a multi-step strategy where they consider some transformation on the data without explaining the main structure, in addition a Normal distribution as a basic assumption and estimate the coefficients b(x) of GLM without mention the estimation method. While GLM and statistical modeling and parameter mapping have been, extensively used in various fields, it would be better- based on my knowledge- to consider the nonlinear view which the main part of the image and movement to make a better predict model.

The manuscript is interesting however, it needs more clarity in mathematical derivations and notations, and it should be, compared to quite robust statistical methods to make the novelty clearer.

English language of manuscript also must be improved

6. PLOS authors have the option to publish the peer review history of their article (what does this mean?). If published, this will include your full peer review and any attached files.

Reviewer #1: No

Reviewer #2: No

---

## [Author Response · Author response to Decision Letter 0]

26 Jan 2020

We thank the reviewers and the senior editor for communicating their feedback on our manuscript. We have revised the manuscript by addressing the concerns raised as best as we

could. In situations where we could not fully address a reviewer’s concern with additions to the manuscript, we provide a detailed explanation as to why we are limited in our ability to do so. Due to the extent of the changes required, and the need to revise and include figures in our response, we have chosen to attach a PDF containing a full response to all reviewer comments instead of inputting them into the "Respond to Reviewers" form. We invite the reviewers and the editor to read the attached response letter and we hope it meets their expectations.

The revisions suggested by the reviewers have further strengthened our manuscript and we look forward to your favorable consideration.

Sincerely,

Brian G. Booth, Eva Hoefnagels, Toon Huysmans, Jan Sijbers, and Noël L.W. Keijsers (the authors)

---

## [Decision Letter · Decision Letter 1]

12 Feb 2020

PAPPI: Personalized analysis of plantar pressure images using statistical modelling and parametric mapping

PONE-D-19-27376R1

Dear Dr. Booth,

We are pleased to inform you that your manuscript has been judged scientifically suitable for publication and will be formally accepted for publication once it complies with all outstanding technical requirements.

With kind regards,

Manoj Srinivasan, Ph.D.

Academic Editor

PLOS ONE

Additional Editor Comments (optional):

Reviewers' comments:

Reviewer's Responses to Questions

**Comments to the Author**

1. If the authors have adequately addressed your comments raised in a previous round of review and you feel that this manuscript is now acceptable for publication, you may indicate that here to bypass the “Comments to the Author” section, enter your conflict of interest statement in the “Confidential to Editor” section, and submit your "Accept" recommendation.

Reviewer #1: All comments have been addressed

2. Is the manuscript technically sound, and do the data support the conclusions?

Reviewer #1: Yes

3. Has the statistical analysis been performed appropriately and rigorously? 

Reviewer #1: Yes

4. Have the authors made all data underlying the findings in their manuscript fully available?

Reviewer #1: Yes

5. Is the manuscript presented in an intelligible fashion and written in standard English?

Reviewer #1: Yes

6. Review Comments to the Author

Reviewer #1: The authors did a great job in answering all reviewer comments. I have no further edits nor comments to propose. The paper is an excellent contribution to the literature.

7. PLOS authors have the option to publish the peer review history of their article (what does this mean?). If published, this will include your full peer review and any attached files.

Reviewer #1: No

---

## [Editor Report · Acceptance letter]

13 Feb 2020

PONE-D-19-27376R1 

PAPPI: Personalized analysis of plantar pressure images using statistical modelling and parametric mapping 

Dear Dr. Booth:

I am pleased to inform you that your manuscript has been deemed suitable for publication in PLOS ONE. Congratulations! Your manuscript is now with our production department. 

With kind regards,

on behalf of

Dr. Manoj Srinivasan 

Academic Editor

PLOS ONE